
**Evaluation of human risks of surface and groundwater contaminated with**
**Cd and Pb south of El-Minya Governorate, Egypt.**
Salman Salman[1], Ahmed A. Asmoay[1], Amr El-Gohary[1], Hassan Sabet[2]
[1] Geological Sciences Dep. National Research Center, Dokki, Giza, 12622, Egypt.
[2] Geology Dep., Faculty of Science, Al-Azhar University, Cairo, Egypt.
*Correspondence to*: Ahmed A. Asmoay (asmoay@gmail.com)
**Abstract:**
Water pollution with Cd and Pb has worldwide concern because of their health impact.
Evaluation of their concentrations and potential human health risks of surface and
groundwater south El-Minya Governorate, Egypt is the main aim of the study. Fifty-five
samples were collected; 30 samples surface water and 25 samples groundwater. The samples
were analyzed using Atomic Absorption Spectrometry (AAS) to determine Cd and Pb
contents. The heavy metals levels in both of surface and groundwater exceeded the maximum
allowable level for drinking water which set by WHO. The hazard quotient and hazard index
showed that groundwater may pose a health risk to residents, especially the children,
primarily due to the high Cd content. In addition, there might be some concern for adverse
Carcinogenic health effects. The pollution returns to human activities. The water can be
recommended for irrigation not for drinking.
**1. Introduction**
Water pollution resources pollution was becoming a worldwide problem. To protect
the environment and public health, it is important to have precise knowledge on amount and
type of water pollutants especially heavy metals. Because the heavy metals have long
biological half-life, they are threat the human health in case of excessive content (Albji et al.,



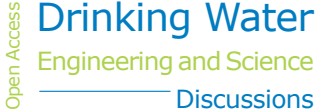

2013). Cadmium and lead are of the most chemical pollutants that threaten the water quality for different uses. Because of their harmful effects, persistence, identification and monitoring of their concentrations in water are of critical importance for protecting ecological and human health (Osei et. al., 2010; Nazar, et. al., 2012).

Cadmium has a major environmental concern and ranked as the sixth significant human health hazard toxic substances (ATSDR, 1997). It is released into the aqueous system from metal plating, smelting, mining, cadmium-nickel batteries, phosphate fertilizers, paint industries, pigments and alloy industries as well as from sewage (Kadirvalu and Namasivayam, 2003). The nervous system appears to be the most sensitive target of Cd toxicity. Cadmium exposure can produce a wide variety of acute and chronic effects in humans such as renal failure, lung insufficiency, bone lesions and hypertension (Gupta and Bhattacharyya, 2007; Sun and Li, 2011).

Lead is used in many industries including lead smelting and processing, batteries manufacture, pigments, solder, plastics, cable sheathing, ammunition and ceramics. It was the most common environmental contaminant (Chiang et. al, 2012; Fischbein, 1998). Water and soil contaminated with lead pose serious human health risks with global dimensions (Tong et. al., 2000; Brooks et. al., 2010). Lead does not undergo degradation or decomposition. Thus, its long persistence in the environment exacerbates its threat to human health. Lead absorbed by human body disturbs many body processes and is harmful to many organs and tissues such as heart, bones and nervous systems (Needleman, 2004; Bruce et. al., 2012). Fumes from lead-based paints, automobile exhaust, polluted air of industrial plants and cigarette smoke may all contain lead, therefore, products containing lead are now prohibited (Moreira and Moreira, 2004). Due to urbanization, lead and other metals are regularly discharge into fields, water and soils through sewage sludge (Abreu et al., 1998).



Depending on their concentration, heavy metals can result in a wide range of toxic
effects on humans, plants, animals, and microbes (Caliza et al., 2012; Qu et al., 2012).
Quantify both of carcinogenic risks for Cd and non-carcinogenic risk for Cd and Pb to
children and adults is important. Human risk assessment methodologies are well developed
and documented in lots of investigations by taking into consideration exposure scenarios of
metal intake through contaminated water (Muhammad et al., 2011; Shah et al., 2012; Dou and
Li, 2012). The hazard quotients (HQ) of the USEPA (1989) are extensively used to
characterize the non-carcinogenic health effects of toxic metals by comparison of their
exposure effects to a reference dose (R$f$D) (Qu et al., 2012).
The objective of the present study is to investigate the distribution of cadmium and
lead in the surface and ground water systems in the western part of the River Nile between
Abu Qurqas and Dyer Mawas districts, El-Minya Governorate, Egypt.
**1.1. Location**
The study area occupied the middle part of the Nile Valley between longitudes 30° 29′
and 30° 54′E and latitudes 27° 37′and 27° 56′N (Fig. 1). It is bounded by the River Nile from
the east and the calcareous plateau at the west between Abu Qurqas northward and Dyer
Mawas at the south. The water resources in the study area are represented by the River Nile,
canals and drains as well as groundwater (Fig. 1). The River Nile passes through high eastern
and western calcareous plateaus with a general slope from south to north about 0.1 m/km
(Korany et al 2006). The stratigraphic succession in El-Minya area is represented by Tertiary
and Quaternary sedimentary rocks (Fig. 2). The distribution of the different rock units was
indicated in Said (1981). The stratigraphic sequence is built up from base to top as follow:
Middle Eocene limestone intercalated with shale (Samalut Formation); Pliocene
undifferentiated sands, clays, and conglomerates; Plio-Pleistocene sand and gravel with clay



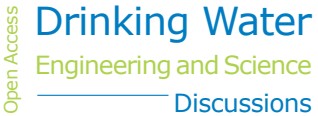

and shale lenses; Pleistocene sand and gravel with clay lenses and Holocene silt and clay. The
main aquifer in the study area is represented by Pleistocene sediments which compose of sand
and gravel of different sizes with some clay intercalation. The thickness of this aquifer ranged
from 25 to 300 m from desert fringes to central Nile Valley (Sadek 2001). The aquifer is
underlined by impermeable Pliocene clay layer and overlain by semi-permeable silty clay
layer. The semi-confined bed (silty clay) is missed outside the floodplain and the aquifer
becomes unconfined westward in the desert fringes. The groundwater flows generally from
the southern part to the northern part of the study area. Locally, the groundwater flows from
the center outwards in all directions; therefore, the River Nile is a recharge zone. The aquifer
is recharged by Nile water, irrigation system, drains, agricultural infiltration and vertical
upward from the deeper saline aquifers (Korany, 1984).
## 2.  Material and methods
In November 2014, thirty water samples were collected from surface water resources
at the study area (Fig. 1 and Table. 1). In addition, 25 groundwater samples were collected
from the Quaternary aquifer (Fig. 1). Pre-rinsed polypropylene bottles were filled with the
samples, sealed tightly. At lab the samples were filtered through filter paper (Whatman No.
42) and digested with nitric acid (APHA, 1995). Samples were analyzed using atomic
absorption spectrometer instrument (model: Perkin Elmer 400) in National Research Centre
Laboratories.
For health risk assessment, two major indices, chronic daily intake (CDI, mg/kg/day)
and hazard quotient (HQ) for each contaminant was calculated according to the following
equations (Eq. 1, 2, 3) adopted by Kelepertzis (2014):

$$CDI = C*IR*ED*EF/BW*AT \qquad (1)$$

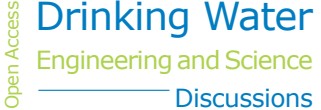



$$\text{HQ}_{\text{non-carcinogenic}} = \text{CDI} / \text{R}f\text{D} \qquad\qquad (2)$$

$$\text{HQ}_{\text{carcinogenic}} = \text{CDI} *\text{SF} \qquad\qquad (3)$$

where, C, IR, ED, EF, BW, AT and R$f$D represent the concentration of metal in water (mg/L), average daily intake rate (2 L/day for adult and 1.2 L/day for children), exposure duration (15 years), exposure frequency (350 days), body weight (70kg for adult and 28kg for children), average time (ED*365 days for non-carcinogenic and lifetime*365 for carcinogenic risk) and toxicity reference dose. According to USEPA (2011) the R$f$D for Cd and Pb is0.0005 and 0.055 mg/kg/day respectively. Also, slope factor (SF) for Cd and Pb is 0.38 and 0.055 mg/kg/day respectively. The average life time for adult is 65 years and 6.5 years for children.

## 3. Results and discussion

### 3.1. Surface water

River Nile and its tributaries (canals and drains) are the main source of water in Egypt especially for the governorates allocated on the river banks and on its branches. Therefore, the quality of water was evaluated by measuring of Cd and Pb concentrations in south of El-Minya Governorate during 2014 (Table 2). The close difference between mean and median (Table 2) indicate the homogenous distribution of both metals and the unity of their source. Cadmium concentrations (Table 2), ranged from 1 to 48µg/l, exceed the permissible limit (3µg/l) for drinking water according to WHO (2011). Excess Cd could accumulate in the kidney and remains for many years causes irreversible kidney damage (Goyer, 1996). The kidney patients in the study area is presumed to have increased from 10 patient/million in 1974 to about 165 patient/million in 1995 and in 2005 it was 260 patient/million in El-Minya Governorate (El Minshawy and Kamel, 2006). Agricultural activities are considered as the most sources for Cd, where the Egyptian marine phosphorite used for manufacture of super-

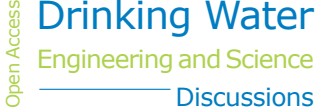

phosphate fertilizers contains up to 20 ppm Cd (El-Kammar, 1974). Pesticides also can lead to
high Cd content in the study area (Bowen, 1966).
The highest concentration of Cd was recorded in sample number $S_8$ of 48μg/l close to
Abu Qurqas Sugar factory due to throwing of human wastes or garbage (Al-Shiekh Sharf
canal). The lower concentration was recorded in sample number $S_6$ (1μg/l) which was picked
up from the River Nile. These results are in line with those obtained Toufeek (2011) who
recorded average Cd concentration 12.5 μg/l at Aswan (southern Egypt), while Salman (2013)
found that the Cd level in samples collected from Sohag Governorate were flocculated around
16μg/l. Therefore, Osman and Kloas (2010) mentioned that the average Cd concentration in
the River Nile at Assuit was 6 μg/l.
Lead concentration ranged from 54 to 329μg/l in the studied surface water samples
(Table. 2). The study samples content of Pb passes the permissible limit (10μg/l) for drinking
water according to WHO (2011). Lead adsorbed by human body disturbs many body
processes and is harmful to many organs and tissues such as heart, bones, nervous system
(Needleman, 2004; Bruce et al., 2012). The highest concentration Of Pb was recorded in
sample number $S_{28}$ (329μg/l) from Al-Nasriyah canal, while sample number $S_6$, which was
collected from the River Nile at Abu Qurqas district, contains the lowest concentration
(54μg/l). These results are in agreement with those obtained by Toufeek (2011) who reported
about 214μg/l Pb in the River Nile at Aswan. In addition, Osman and Kloas (2010) proved
that the average Pb concentration in the River Nile at Assuit is nearly 24μg/l.
The Nile River and canals contain higher Cd and Pb concentrations than the drains
indicating the role of the human activities as a main source of these metals because most of
the canals penetrate settlements and adjacent to roads. In addition, most of the houses have
not sewers and discharge their wastewater and rubbish into canals.

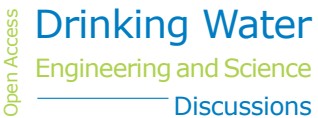

### 3.2. Groundwater

Groundwater is the second important water resources in the study area and the only water resource in the desert fringes. It is used for irrigation and unfortunately for domestic and drinking in some villages which lacking safe potable water source. The groundwater samples exhibit relative wide range of the Cd level varying from 2 to 49μg/l with mean value 24μg/l (Table 3). The groundwater Cd level decreased eastward (Fig. 3) due to mixing with the surface water from River Nile and the role of the silty clay layer in the absorbance of Cd and prevent it to reach the aquifers. However, there are three hot spots of Cd resulted from the intensive human activities and fuel stations. Cd hot spot in the NW part of the study area adjacent to the western desert road was developed as a result of the unconfined condition of the aquifers which become vulnerable to contamination.

Also, the measured Pb content of the analyzed groundwater samples show relative wide range varying from 90 to 410μg/l with an average of 242μg/l (Table. 3). The marked high level of Pb content implies that the anthropogenic activities are the main source of Pb. The results are in agreement with Melegy et al (2014) who mentioned that Cd and Pb concentration in the groundwater of Sohag were around 21 and 383 μg/l, respectively. In addition, Salman (2013) reported that the average of Cd and Pb concentrations are 21 and 383μg/l respectively in the Quaternary aquifer at Sohag. Pb level in the groundwater of the study area was increased at Abu Qurqas district resulting from the effect of the sugar factory, cesspits and fuel stations (Fig. 4).

All the samples are unsuitable for drinking purpose where they possess Cd and Pb values above the permissible limit of 3μg/l for Cd and 15 μg/l for Pb (WHO, 2011). On the other hand, surface and ground water are suitable for irrigation purposes according to NAS-NAE (1973), where they contain less than 10000 and 5000000μg/l of Cd and Pb, respectively. But unfortunately, using the groundwater for drinking in some villages of the study area



represents serious health impact. El-Minshawy and Kamel (2006) mentioned that the use of
unsafe water for drinking contributes up to 71.8% of the renal failure in the study area.
**3.3. Health risk assessment**
It was observed that general population in the rural area is using groundwater from
hand pumps for drinking and domestic purposes because they don't have access to the tap
water from the tube wells.  Therefore, health risk assessment for groundwater was carried out
in this study. The results of non-carcinogenic and carcinogenic health risks due to metal
exposure in groundwater samples are provided in Table (4).
The values of non-carcinogenic health risk of Cd in drinking water for adults range
from 0.11 to 2.68 with an average of 1.29 and for children extend between 0.16 and 4.03
averaging 1.94 (Table 4).  The Pb values for adults range from 0.04 to 0.20 with an average of
0.12 and for children vary from 0.07 to 0.31 averaging 0.18. According to USEPA (2011), Cd
and Pb values of non-carcinogenic health risk should not exceed 1 to be considered as non-
harmful drinking water.
The Cd values of carcinogenic health risk for drinking water was ranged from $4.8*10^{-6}$
to $1.2*10^{-4}$ averaging $5.66*10^{-5}$ for adults and from $2.9*10^{-5}$ to $7.11*10^{-4}$ with an average of
$3.42*10^{-4}$ for children. Pb values for adults extend between $3.13*10^{-5}$ and $14.3*10^{-5}$ with an
average of $8.41*10^{-5}$, while the results for children vary from $1.67*10^{-9}$ to $86.1*10^{-5}$ averaging
$48.5*10^{-5}$ (Tables 4). According to USEPA (2011), the values of Cd and Pb of carcinogenic
health risk should not exceed $10^{-6}$ which it is the safe limit of the hazard index. More than this
limit, the drinking water has very harmful effects on the inhabitants.

**4.  Conclusion**
Cadmium and lead contents of the studied samples from Nile River, its tributaries
(canals and drains) and groundwater exceed the permissible limits for drinking water and

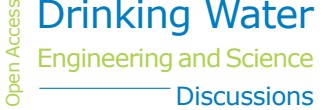



could be disturbing many body processes and are harmful to many organs and tissues such as the heart, bones, kidney and nervous system. The canals are higher in Cd and Pb concentrations than the drains indicating the role of the human activities as a main source of these metals because most of the canals penetrate settlements and adjacent to roads. The lowest concentrations of Cd and Pb were recorded in samples which were picked up from the River Nile.

The non-carcinogenic health risk of Cd values exceeds 1 which is the safe limit of the hazard index indicating harmful drinking water while the Pb values do not pass that limit. The values of Cd and Pb of carcinogenic health risk exceed the safe limit of the hazard index and accordingly the drinking water has very harmful effects on the inhabitants.

The water resources in the study area (surface and groundwater) are suitable for irrigation purposes. Source of pollution in the investigated area were derived from anthropogenic activities such as industries, agriculture, mining and sewage. The water in concerned area is suitable to use for irrigation purpose and unsuitable for drinking.

**Recommendations**

It is recommended to connect the houses in different rural parts of the study area with safe drinking water lines with regular monitoring of water resources and the end user water lines.

**Acknowledgments**

The authors gratefully acknowledge the National Research Centre for funding this research as a Ph.D. internal project and the grant no. is (8/5/9) to support Mr. Ahmed A. Asmoay to do the lab work.

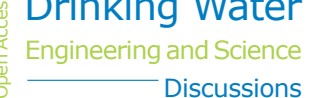

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

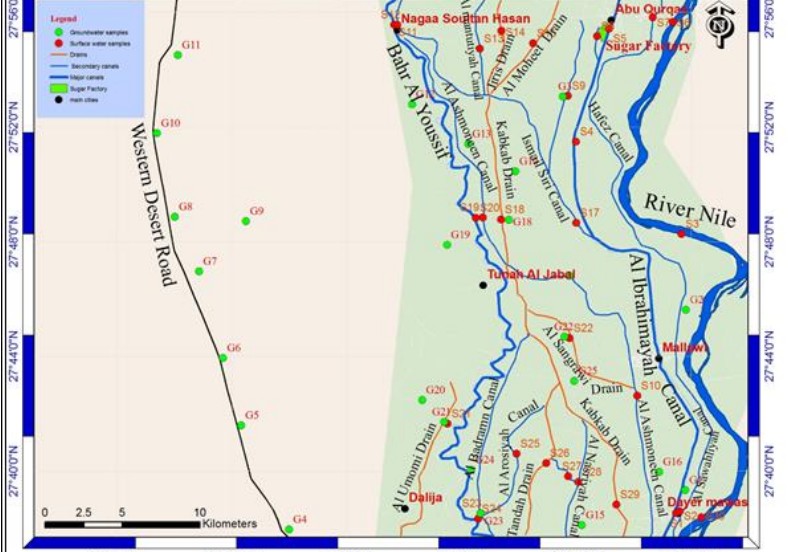

8        **Figure 1: Location map of the study area.**

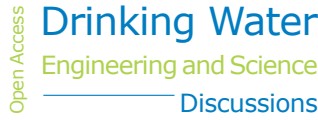

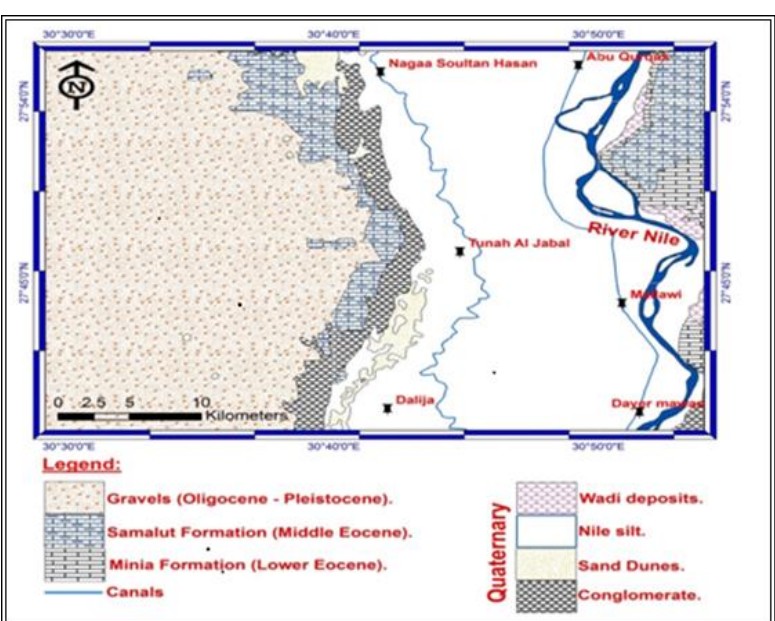

**Figure 2: Geologic map of the study area.**

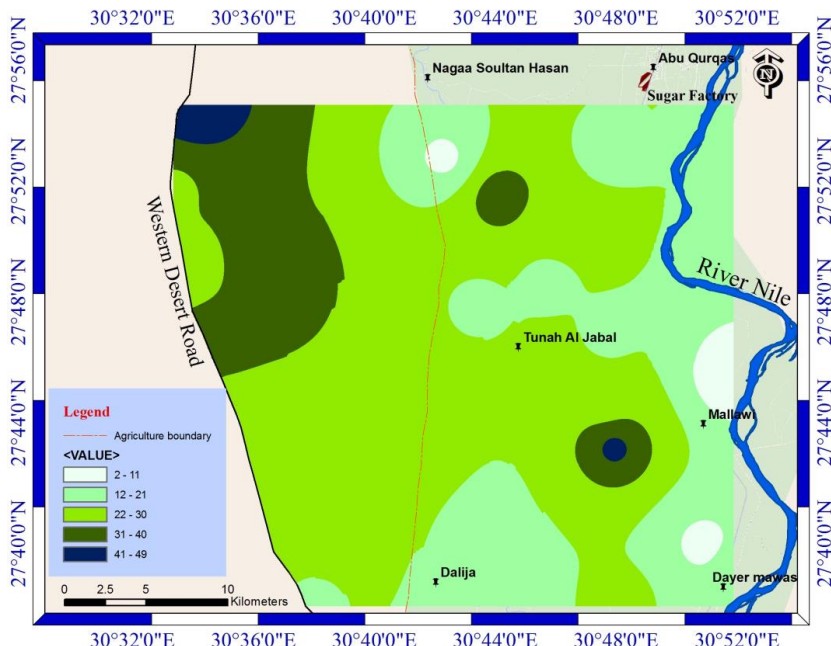

**Figure 3: Spatial distribution map of Cd in the studied groundwater samples.**

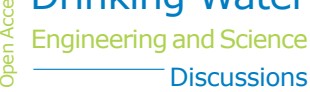

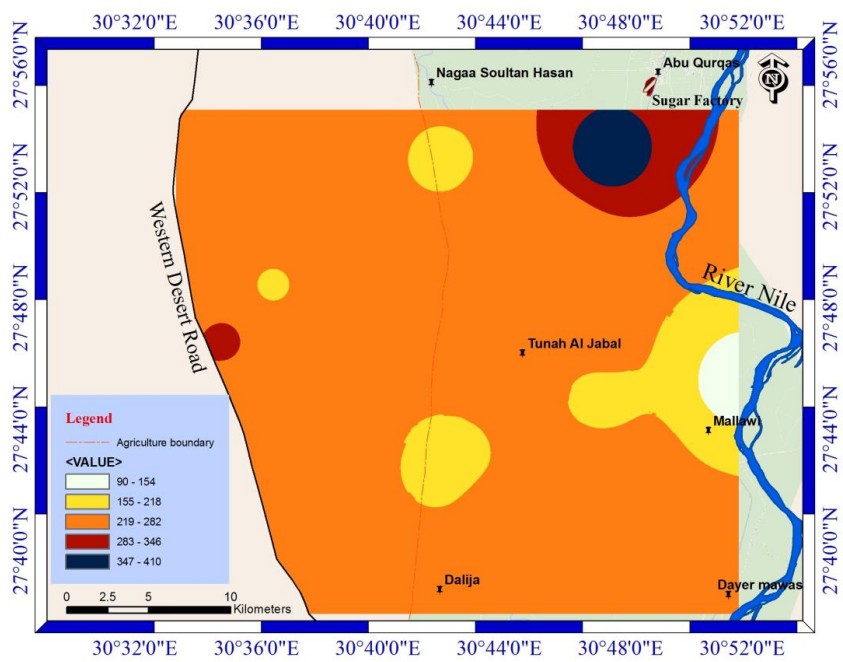

2        **Figure 4: Spatial distribution map of Pb in the studied groundwater samples.**

4        **Table 1: Surface water samples and localities.**

| Sample No. | Canals | Sample No. | Canals & Drains |
|---|---|---|---|
| $S_3$, $S_6$, $S_{30}$ | River Nile | $S_{28}$ | Al Nasriyah canal |
| $S_1$, $S_5$ | Al Ibrahimayah canals | $S_9$ | Branch from Ismail Siri canal |
| $S_{11}$, $S_{20}$, $S_{24}$ | Bahr Youssef | $S_{27}$ | Branch from Al Nasriyah canal |
| $S_2$ | Al Sawahliyah canals | $S_4$ | Al Sellic Drain |
| $S_7$ | Hafez canal | $S_{14}$ | Jiris Drain |
| $S_8$ | Al ShiekhSharf canal | $S_{15}$ | Al Moheet  Drain |
| $S_{10}$, $S_{16}$, $S_{19}$ | Al Ashmoneen canal | $S_{18}$, $S_{29}$ | Kabkab Drain |
| $S_{12}$, $S_{17}$ | Ismail Siri canal | $S_{21}$ | Al umomi Drain |
| $S_{13}$ | Al Mantutiyah canal | $S_{22}$ | Al Sangrawi Drain |
| $S_{23}$ | Al Badraman canal | $S_{26}$ | Tandah Drain |
| $S_{25}$ | Al Arosiyah canal | | |



1    **Table 2:  Cd and Pb concentrations in the surface water samples (µg/l).**

| Sample No. | Nile & Canals | | Drains | | Sample No. | Nile & Canals | | Drains | |
|---|---|---|---|---|---|---|---|---|---|
| | Cd | Pb | Cd | Pb | | Cd | Pb | Cd | Pb |
| $S_1$ | 39 | 184 | - | - | $S_{18}$ | - | - | 17 | 216 |
| $S_2$ | 37 | 210 | - | - | $S_{19}$ | 15 | 208 | - | - |
| $S_3$ | 22 | 270 | - | - | $S_{20}$ | 15 | 208 | - | - |
| $S_4$ | - | - | 22 | 222 | $S_{21}$ | - | - | 28 | 183 |
| $S_5$ | 19 | 241 | - | - | $S_{22}$ | - | - | 21 | 260 |
| $S_6$ | 1 | 54 | - | - | $S_{23}$ | 33 | 275 | - | - |
| $S_7$ | 19 | 96 | - | - | $S_{24}$ | 21 | 250 | - | - |
| $S_8$ | 48 | 288 | - | - | $S_{25}$ | 7 | 272 | - | - |
| $S_9$ | 9 | 209 | - | - | $S_{26}$ | - | - | 22 | 273 |
| $S_{10}$ | 20 | 222 | - | - | $S_{27}$ | 16 | 298 | - | - |
| $S_{11}$ | 19 | 268 | - | - | $S_{28}$ | 35 | 329 | - | - |
| $S_{12}$ | 43 | 262 | - | - | $S_{29}$ | - | - | 23 | 198 |
| $S_{13}$ | 28 | 303 | - | - | $S_{30}$ | 28 | 192 | - | - |
| $S_{14}$ | - | - | 42 | 234 | Mean | 24 | 233 | 23 | 224 |
| $S_{15}$ | - | - | 12 | 213 | Median | 21 | 245 | 22 | 219 |
| $S_{16}$ | 32 | 225 | - | - | Min. | 1 | 54 | 12 | 183 |
| $S_{17}$ | 22 | 261 | - | - | Max. | 48 | 329 | 42 | 273 |

3    **Table 3:  Cd and Pb concentrations (µg/l) in the study groundwater samples.**

| Sample No. | Cd | Pb | Sample No. | Cd | Pb |
|---|---|---|---|---|---|
| $G_1$ | 15 | 249 | $G_{15}$ | 24 | 274 |
| $G_2$ | 2 | 90 | $G_{16}$ | 8 | 242 |
| $G_3$ | 19 | 410 | $G_{17}$ | 19 | 228 |
| $G_4$ | 20 | 224 | $G_{18}$ | 19 | 228 |
| $G_5$ | 26 | 252 | $G_{19}$ | 19 | 243 |



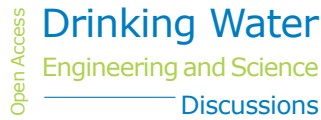

| $G_6$ | 23 | 242 | $G_{20}$ | 28 | 162 |
|---|---|---|---|---|---|
| $G_7$ | 36 | 296 | $G_{21}$ | 14 | 266 |
| $G_8$ | 25 | 230 | $G_{22}$ | 27 | 193 |
| $G_9$ | 39 | 213 | $G_{23}$ | 17 | 275 |
| $G_{10}$ | 29 | 229 | $G_{24}$ | 19 | 264 |
| $G_{11}$ | 49 | 254 | $G_{25}$ | 42 | 276 |
| $G_{12}$ | 9 | 197 | Mean | 24 | 242 |
| $G_{13}$ | 38 | 270 | Minimum | 2 | 90 |
| $G_{14}$ | 23 | 242 | Maximum | 49 | 410 |

**Table 4: Statistical parameters of non-carcinogenic and carcinogenic health risks for**
**groundwater samples.**

| Parameter | Non-carcinogenic for Adults | | Non-carcinogenic for Children | |
|---|---|---|---|---|
| | HQ Cd | HQ Pb | HQ Cd | HQ Pb |
| **Minimum** | 0.11 | 0.04 | 0.16 | 0.07 |
| **Maximum** | 2.68 | 0.20 | 4.03 | 0.31 |
| **Average** | 1.29 | 0.12 | 1.94 | 0.18 |
| | Carcinogenic for Adults | | Carcinogenic for Children | |
| | HQ Cd | HQ Pb | HQ Cd | HQ Pb |
| **Minimum** | $4.8*10^{-6}$ | $3.13*10^{-5}$ | $2.9*10^{-5}$ | $1.67*10^{-9}$ |
| **Maximum** | $1.2*10^{-4}$ | $14.3*10^{-5}$ | $7.11*10^{-4}$ | $86.1*10^{-5}$ |
| **Average** | $5.66*10^{-5}$ | 8.41E-05 | $3.42*10^{-4}$ | $48.5*10^{-5}$ |

HQ = Hazard quotient; HQ $_{\text{non-carcinogenic}}$ = CDI / R$f$D
HQ $_{\text{carcinogenic}}$ = CDI / SF **Table 4:** Statistical parameters of non-carcinogenic and carcinogenic
health risks for groundwater samples.

| Parameter | Non-carcinogenic for Adults | | Non-carcinogenic for Children | |
|---|---|---|---|---|
| | HQ Cd | HQ Pb | HQ Cd | HQ Pb |
| **Minimum** | 0.11 | 0.04 | 0.16 | 0.07 |
| **Maximum** | 2.68 | 0.20 | 4.03 | 0.31 |
| **Average** | 1.29 | 0.12 | 1.94 | 0.18 |
| | Carcinogenic for Adults | | Carcinogenic for Children | |



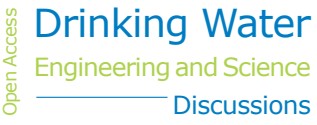

|            | HQ Cd          | HQ Pb           | HQ Cd           | HQ Pb            |
|------------|----------------|-----------------|-----------------|------------------|
| **Minimum** | $4.8*10^{-6}$  | $3.13*10^{-5}$  | $2.9*10^{-5}$   | $1.67*10^{-9}$   |
| **Maximum** | $1.2*10^{-4}$  | $14.3*10^{-5}$  | $7.11*10^{-4}$  | $86.1*10^{-5}$   |
| **Average** | $5.66*10^{-5}$ | 8.41E-05        | $3.42*10^{-4}$  | $48.5*10^{-5}$   |

1       HQ = Hazard quotient; HQ $_{non\text{-}carcinogenic}$ = CDI / R$f$D

2       HQ $_{carcinogenic}$ = CDI / SF