# Peer review of "Evaluation of human risks of surface and groundwater contaminated with Cd and Pb south of El-Minya Governorate, Egypt."

_Drinking Water Engineering and Science, 2018_

## Referee Comment (RC1) · Anonymous Referee #1 · 14 Jan 2019

This paper deals with monitoring of Cd and Pb levels in surface and groundwater of area in between El-Minya and Assuit Governorates, Egypt. The hazard quotient for each contaminant was calculated for investigating the health risk assessment. There is a lack of novelty and the risk of Cd and Pb in the surface water was ignored, while it was mentioned in the aim. Therefore, this paper could not publish in Drink. Water Eng. Sci. journal in the current quality. This decision is according to the following comments: (1) The aim of this study in the title and in the Abstract is for evaluation of human risks of surface and groundwater contaminated with Cd and Pb. The value of this risk is not emphasized, as it was calculated only for the groundwater that be used with unidentified inhabitants and the treatment of the surface water did not put

into account.

(2) In the Abstract, authors referred to the hazard index (the summation of HQ) (Page 1, Line 15), while there is no information in the whole manuscript about this parameter and how they calculate it.

(3) In the Introduction: # (Page 3, Line 10), the aim should be the same as written in the Abstract. Also, authors did not show the novelty of the work. What is done before in risk assessment research work? # (Page 3, Line 22 until the end of the paragraph) the description of the nature of rocks and stones in the area did not connected with the presence of the Cd and Pb problem.

(4) Materials and Methods: # (Page 4, Line 14) the samples were taken in November 2014. The results must be up-to-date, unless the authors mention that there is no action or changing for the situation yet. # Authors mentioned that they filtered the collected samples and then digested them with nitric acid. This is give the concentration of heavy metals in the water filtrate (dissolved heavy metals). The samples should acidified first or digested and then filtered to have the total and exact heavy metal concentration. Also, the standard method that they refer for this analysis is not up-to-date. APHA, 2017 is recommended. # (Page 5, Line 5), the exposure frequency is 350 day/year. # (Page 5, Line 6), how much lifetime is used for the calculations? # (Equation 3), the slope factor (SF) is not defined and the reference for SF values used for Cd and Pb is missed.

(5) Results and Discussion # (Page 6, Line 5), Why the Cd concentrations in the River Nile (S6, S7) are lower than that for River Nile (S2, S3)?. While S2 and S3 are from the southern part and even S6 and S7 sites are near to Abu Qurqas Sugar Factory. Authors claim that this Sugar Factory is a source of Cd pollutants. # Authors did not explain the source of Pb in the River Nile, especially at S2 and S3 sites. # S4 appears in the map (Fig. 1) as a canal site and written in the Table 2 as a drain site. # Authors refer in Page 5, Line 17 to the unity of Cd and Pb source and they mentioned in the same page 5,

Line 23 that the agricultural activities (Fertilizers and pesticides) are the main sources for Cd. Then they mentioned in Page 6 Line 21 that the Cd and Pb concentrations in the River Nile and canals are higher than that in the drains. They explained that due to the high human activities. Actually, what is the main source of Cd and Pb pollution? # What is the source of Pb in the River Nile samples (S2&S3)? # The area under study is covered by drinking water distribution network. There is no information about that. In Page 7, Line 4, how many inhabitants not covered by the DW network and safe potable water? And how they are far from the distribution system? # Why the levels of Cd and Pb in the groundwater sites near to the Western Deseret Road are close to that in River Nile while these wells are far from the River by about 25 Km and far from the canal (Bahr El Youssif) by about 15 Km. # The health risk assessment in this research article is based only on the exposure of some inhabitants in some villages on the study area with Cd and Pb via ingestion of contaminated groundwater. The study ignored the contact with the surface water in irrigation activities and did not give full picture for the potable water resources, especially that there are water treatment facilities covering the area. # The resolution of maps is so low.

(6) The number of references (40) is so high for research article.

---

## Short Comment (SC1) · 19 Jan 2019

Dear Editor and Authors

It is my pleasure to write a comment about the paper entitled: Evaluation of human risks of surface and groundwater contaminated with Cd and Pb south of El-Minya Governorate, Egypt.

The research dealt with the assessment of pollution in the study area in a new way with HQ (Hazard Quotient). The paper is well written.

Only one comment about CDI (chronic daily intake), the authors mentioned in (Material

and Methods section) page 4 rows 21 and 22 that they used (two major indices, chronic daily intake (CDI, mg/kg/day) and hazard quotient (HQ)), I think it is only one Index HQ and authors used CDI to calculated HQ.

With my best regards Dr. Ahmed Elnazer
* * *

---

## Referee Comment (RC2) · Anonymous Referee #2 · 21 Jan 2019

In this paper Cd and Pb are monitored in groundwater and surface water and compared to WHO guidelines. The paper is poorly written and not very innovative, since most of the sampling was already done before. General comments: - Give clear objective (and knowledge gap) at the end of the introduction. - "Location" should be part of the Materials and Methods section (not of introduction) - Explain why Health Risk Assessment gives other information than WHO guidelines - Check language, including tenses. - Introduce abbreviations (like Cd and Pb) once and then use the abbreviations in the rest of the manuscript. - Structure the description of Cd and Pb in the introduction in the same way. - Use same structure for describing Cd and Pb (as on pg 2) - Avoid repetitions in the paper (like first 4 sentences on pg 3). Specific comments: - Pg 1, line

17, delete "in addition,. . . human activities" from abstract. - Pg1, line 21, 2x "pollutions" - Pg 1, line 22, amount = concentrations - Pg 1, line 24, content = concentrations - Pg 2, line 19, delete second "body" - Pg 5, line 16-17, delete sentence "the close. . .. their source". - Pg 5, line 19-20, not relevant - Pg 5, line 24, insert "important" between "most" and "source" - Pg 6, line 4, "throwing" = "deposition" - Pg 6, line 5, "picked" = "collected" - Pg 6, line 13-15, not relevant here. - Pg 6, line 24, "rubbish" = "solid waste". - Pg 7, line 2, "second water source". . . "and the only one in the desert fringes" - Pg 7, line 3, delete "unfortunately" - Pg 7, line 4, delete "which lacking safe potable water source". - Pg 7, line 7, "absorbance" = "adsorption" - Pg 7, line 10, ". . .desert road, is vulnerable as a result of the unconfined condition of the aquifers". - Pg 7, line 14, which type of anthropogenic activities? - Pg 7, line 24, seem to be very high values.. - Pg 9, therefore treatment is needed before water containing Cd and Pb can be used for drinking

---

## Author Comment (AC1) · 21 Jan 2019

Dear editor, we are very grateful for those worthy comments, we would you like to publish our article in this valuable journal. We taken these comments in our consideration and the response as the following:- 1) Indeed, the risk health of surface water and groundwater samples (HQ) was calculated based on the chronic daily intake (CDI) as mentioned equations in page 4 line 25 and page 5 lines 1and 2. Last but not least we hope that our article to be published in your valuable journal.

Please also note the supplement to this comment:

[Figure]

https://www.drink-water-eng-sci-discuss.net/dwes-2018-37/dwes-2018-37-AC1-supplement.pdf

[Figure]

**Supplement:**

[revised manuscript text omitted]
 (HQ) due to metal exposure in surface water and groundwater samples are provided in Tables (4) and (5).

The non-carcinogenic health risk values for Cd in the drinking surface water for adults vary from 0.05 to 2.06 with an average of 1.30 and for children fluctuate between 0.08 and 3.95 with mean 1.96 (Table 4). The Pb values for adults range from 0.03 to 0.16 with mean 0.11 and for children extend between 0.04 and 0.25 with an average 0.17 (Table 4). While, the values of non-carcinogenic health risk of Cd in the drinking groundwater for adults range from 0.11 to 2.68 with an average of 1.29 and for children extend between 0.16 and 4.03 averaging 1.94 (Table 5). The Pb values for adults range from 0.04 to 0.20 with an average of 0.12 and for children vary from 0.07 to 0.31 averaging 0.18 (Table 5). According to USEPA (2011), Cd and Pb values of non-carcinogenic health risk should not exceed 1 to be considered as non-harmful drinking water.

The carcinogenic health risk values for Cd in the drinking surface water for adults range from $0.17*10^{-4}$ to $7.9*10^{-4}$ with an average of $3.9*10^{-4}$ and for children extend between $0.1*10^{-5}$ and $4.8*10^{-3}$ with mean $2.3*10^{-3}$ (Table 4). The Pb values for adults vary from $6.2*10^{-3}$ to $37*10^{-3}$ with mean $26*10^{-3}$ and for children range from $3.7*10^{-2}$ to $22*10^{-2}$ with an average $15*10^{-2}$ (Table 4). As well, 
[revised manuscript text omitted]

[Figure]

2          **Figure 1: Location map of the study area.**

[Figure]

2        **Figure 2: Geologic map of the study area.**

[Figure]

4        **Figure 3: Spatial distribution map of Cd in the studied groundwater samples.**

[Figure]

**Figure 4: Spatial distribution map of Pb in the studied groundwater samples.**

**Table 1: Surface water samples and localities.**

| Sample No. | Canals | Sample No. | Canals & Drains |
|---|---|---|---|
| $S_3$, $S_6$, $S_{30}$ | River Nile | $S_{28}$ | Al Nasriyah canal |
| $S_1$, $S_5$ | Al Ibrahimayah canals | $S_9$ | Branch from Ismail Siri canal |
| $S_{11}$, $S_{20}$, $S_{24}$ | Bahr Youssef | $S_{27}$ | Branch from Al Nasriyah canal |
| $S_2$ | Al Sawahliyah canals | $S_4$ | Al Sellic Drain |
| $S_7$ | Hafez canal | $S_{14}$ | Jiris Drain |
| $S_8$ | Al ShiekhSharf canal | $S_{15}$ | Al Moheet Drain |
| $S_{10}$, $S_{16}$, $S_{19}$ | Al Ashmoneen canal | $S_{18}$, $S_{29}$ | Kabkab Drain |
| $S_{12}$, $S_{17}$ | Ismail Siri canal | $S_{21}$ | Al umomi Drain |
| $S_{13}$ | Al Mantutiyah canal | $S_{22}$ | Al Sangrawi Drain |
| $S_{23}$ | Al Badraman canal | $S_{26}$ | Tandah Drain |
| $S_{25}$ | Al Arosiyah canal | | |

**Table 2:  Cd and Pb concentrations in the surface water samples (μg/l).**

| Sample No. | Nile & Canals | | Drains | | Sample No. | Nile & Canals | | Drains | |
|---|---|---|---|---|---|---|---|---|---|
| | Cd | Pb | Cd | Pb | | Cd | Pb | Cd | Pb |
| $S_1$ | 39 | 184 | - | - | $S_{18}$ | - | - | 17 | 216 |
| $S_2$ | 37 | 210 | - | - | $S_{19}$ | 15 | 208 | - | - |
| $S_3$ | 22 | 270 | - | - | $S_{20}$ | 15 | 208 | - | - |
| $S_4$ | - | - | 22 | 222 | $S_{21}$ | - | - | 28 | 183 |
| $S_5$ | 19 | 241 | - | - | $S_{22}$ | - | - | 21 | 260 |
| $S_6$ | 1 | 54 | - | - | $S_{23}$ | 33 | 275 | - | - |
| $S_7$ | 19 | 96 | - | - | $S_{24}$ | 21 | 250 | - | - |
| $S_8$ | 48 | 288 | - | - | $S_{25}$ | 7 | 272 | - | - |
| $S_9$ | 9 | 209 | - | - | $S_{26}$ | - | - | 22 | 273 |
| $S_{10}$ | 20 | 222 | - | - | $S_{27}$ | 16 | 298 | - | - |
| $S_{11}$ | 19 | 268 | - | - | $S_{28}$ | 35 | 329 | - | - |
| $S_{12}$ | 43 | 262 | - | - | $S_{29}$ | - | - | 23 | 198 |
| $S_{13}$ | 28 | 303 | - | - | $S_{30}$ | 28 | 192 | - | - |
| $S_{14}$ | - | - | 42 | 234 | Mean | 24 | 233 | 23 | 224 |
| $S_{15}$ | - | - | 12 | 213 | Median | 21 | 245 | 22 | 219 |
| $S_{16}$ | 32 | 225 | - | - | Min. | 1 | 54 | 12 | 183 |
| $S_{17}$ | 22 | 261 | - | - | Max. | 48 | 329 | 42 | 273 |

**Table 3: Cd and Pb concentrations (µg/l) in the study groundwater samples.**

| Sample No. | Cd | Pb | Sample No. | Cd | Pb |
|---|---|---|---|---|---|
| $G_1$ | 15 | 249 | $G_{15}$ | 24 | 274 |
| $G_2$ | 2 | 90 | $G_{16}$ | 8 | 242 |
| $G_3$ | 19 | 410 | $G_{17}$ | 19 | 228 |
| $G_4$ | 20 | 224 | $G_{18}$ | 19 | 228 |
| $G_5$ | 26 | 252 | $G_{19}$ | 19 | 243 |
| $G_6$ | 23 | 242 | $G_{20}$ | 28 | 162 |
| $G_7$ | 36 | 296 | $G_{21}$ | 14 | 266 |
| $G_8$ | 25 | 230 | $G_{22}$ | 27 | 193 |
| $G_9$ | 39 | 213 | $G_{23}$ | 17 | 275 |
| $G_{10}$ | 29 | 229 | $G_{24}$ | 19 | 264 |
| $G_{11}$ | 49 | 254 | $G_{25}$ | 42 | 276 |
| $G_{12}$ | 9 | 197 | Mean | 24 | 242 |
| $G_{13}$ | 38 | 270 | Minimum | 2 | 90 |
| $G_{14}$ | 23 | 242 | Maximum | 49 | 410 |

**Table 4: Statistical parameters of non-carcinogenic and carcinogenic health risks for surface water samples.**

| Parameter | Non-carcinogenic for Adults | | Non-carcinogenic for Children | |
|---|---|---|---|---|
| | HQ Cd | HQ Pb | HQ Cd | HQ Pb |
| **Minimum** | 0.05 | 0.03 | 0.08 | 0.04 |
| **Maximum** | 2.06 | 0.16 | 3.95 | 0.25 |
| **Average** | 1.30 | 0.11 | 1.96 | 0.17 |
| **Parameter** | Carcinogenic for Adults | | Carcinogenic for Children | |
| | HQ Cd | HQ Pb | HQ Cd | HQ Pb |
| **Minimum** | $0.17*10^{-4}$ | $6.2*10^{-3}$ | $0.1*10^{-5}$ | $3.7*10^{-2}$ |
| **Maximum** | $7.9*10^{-4}$ | $37*10^{-3}$ | $4.8*10^{-3}$ | $22*10^{-2}$ |
| **Average** | $3.9*10^{-4}$ | $26*10^{-3}$ | $2.3*10^{-3}$ | $15*10^{-2}$ |

HQ = Hazard quotient;  $HQ_{non\text{-}carcinogenic} = CDI / RfD$;  $HQ_{carcinogenic} = CDI / SF$

**Table 5: Statistical parameters of non-carcinogenic and carcinogenic health risks for groundwater samples.**

| Parameter | Non-carcinogenic for Adults | | Non-carcinogenic for Children | |
|---|---|---|---|---|
| | HQ Cd | HQ Pb | HQ Cd | HQ Pb |
| Minimum | 0.11 | 0.04 | 0.16 | 0.07 |
| Maximum | 2.68 | 0.20 | 4.03 | 0.31 |
| Average | 1.29 | 0.12 | 1.94 | 0.18 |
| Parameter | Carcinogenic for Adults | | Carcinogenic for Children | |
| | HQ Cd | HQ Pb | HQ Cd | HQ Pb |
| Minimum | $4.8 \times 10^{-6}$ | $3.13 \times 10^{-5}$ | $2.9 \times 10^{-5}$ | $1.67 \times 10^{-9}$ |
| Maximum | $1.2 \times 10^{-4}$ | $14.3 \times 10^{-5}$ | $7.11 \times 10^{-4}$ | $86.1 \times 10^{-5}$ |
| Average | $5.66 \times 10^{-5}$ | $8.41 \times 10^{-5}$ | $3.42 \times 10^{-4}$ | $48.5 \times 10^{-5}$ |

HQ = Hazard quotient;  $HQ_{\text{non-carcinogenic}} = CDI / RfD$;  $HQ_{\text{carcinogenic}} = CDI / SF$

---

## Author Comment (AC2) · 22 Jan 2019

The response Dear editor and referee, we are very grateful for those worthy comments, we would you like to publish our article in this valuable journal. We taken these comments in our consideration and the response as the following: - 1) No one apply the risk health on surface water and groundwater samples (HQ) in Egypt we are the first. If you want to prove that work a research. Indeed, many of researcher carried out studies on the study area but each one has a view that is the scientific research. Each editing in the text sentence has been done. Last but not least we hope that our article to be published in your valuable journal.

[Figure]

Please also note the supplement to this comment:
https://www.drink-water-eng-sci-discuss.net/dwes-2018-37/dwes-2018-37-AC2-supplement.pdf

———————————————————

[Figure]

**Supplement:**

[revised manuscript text omitted]
 (HQ) due to metal exposure in surface water and groundwater samples are provided in Tables (4) and (5).

The non-carcinogenic health risk values for Cd in the drinking surface water for adults vary from 0.05 to 2.06 with an average of 1.30 and for children fluctuate between 0.08 and 3.95 with mean 1.96 (Table 4). The Pb values for adults range from 0.03 to 0.16 with mean 0.11 and for children extend between 0.04 and 0.25 with an average 0.17 (Table 4). While, the values of non-carcinogenic health risk of Cd in the drinking groundwater for adults range from 0.11 to 2.68 with an average of 1.29 and for children extend between 0.16 and 4.03 averaging 1.94 (Table 5). The Pb values for adults range from 0.04 to 0.20 with an average of 0.12 and for children vary from 0.07 to 0.31 averaging 0.18 (Table 5). According to USEPA (2011), Cd and Pb values of non-carcinogenic health risk should not exceed 1 to be considered as non-harmful drinking water.

The carcinogenic health risk values for Cd in the drinking surface water for adults range from $0.17*10^{-4}$ to $7.9*10^{-4}$ with an average of $3.9*10^{-4}$ and for children extend between $0.1*10^{-5}$ and $4.8*10^{-3}$ with mean $2.3*10^{-3}$ (Table 4). The Pb values for adults vary from $6.2*10^{-3}$ to $37*10^{-3}$ with mean $26*10^{-3}$ and for children range from $3.7*10^{-2}$ to $22*10^{-2}$ with an average $15*10^{-2}$ (Table 4). As well, 
[revised manuscript text omitted]

---

## Author Comment (AC3) · 22 Jan 2019

The response Dear editor or referee, we are very grateful for those worthy comments, we would you like to publish our article in this valuable journal. We taken these comments in our consideration and the response as the following:- 1) Indeed, the risk health of surface water samples not calculated, so that it is calculated and listed in Table No 4 in our article. 2) The hazard quotient (HQ) was mentioned in the material and method in page 5 lines 1 and 2 in equations and refers it in the health risk assessment in page 8 lines 7 and 8. 3) This work (health risk assessment) is a new in Egypt didn't apply before. Both of Cd and Pb are derived from anthropogenic sources not geogenic source

in the study area. 4) This work is apart of PhD thesis have been awarded so, isn't up to date. For filtered and digested we carried out this method according to APHA (1995) which is available then. We will do digested before filtration according APHA (2017). The life time is the average of human being age is about 65 year for adults and 6.5 year for children. The reference of slope factor is USEPA (2011) is referred it in page 5 line 10. 5) Sample No S7 and S2 were collected from the canals not the River Nile as listed in Table 2. The levels of metals depend on the place of sample where some samples taken near mooring of boats which causes more pollution. Samples No S6 and S7 are collected from the River Nile (S6) and canal (S7) which not connected by the Sugar Factory outlet despite it is located near the Factory. The sources of Cd and Pb are the anthropogenic which is human activities such as drainage agricultures, sewage water, fuel stations, cesspits, garbage, fertilizers, pesticides mining and industrial waste which reached to water resources. Then the source of Cd and Pb is derived from one source is anthropogenic. The majority of study area is rural society lacks to potable water network, so, the inhabitants depend on groundwater, canals and drains for drinking without any treatment. Groundwater recharges mainly from the River Nile, canals and drains so the level of Cd and Pb are homogenous in the surface and groundwater. We have been published two papers about evaluation both of surface water and groundwater for irrigation and industrial purposes. Source of maps is missed so that difficult increasing the resolution. We will attach the manuscript with response. 6) Increasing of references give scientific motivation for the article.

Last but not least we hope that our article to be published in your valuable journal.

Please also note the supplement to this comment:
https://www.drink-water-eng-sci-discuss.net/dwes-2018-37/dwes-2018-37-AC3-supplement.pdf

————————————————————

[Figure]

**Supplement:**

[revised manuscript text omitted]
 (HQ) due to metal exposure in surface water and groundwater samples are provided in Tables (4) and (5).

The non-carcinogenic health risk values for Cd in the drinking surface water for adults vary from 0.05 to 2.06 with an average of 1.30 and for children fluctuate between 0.08 and 3.95 with mean 1.96 (Table 4). The Pb values for adults range from 0.03 to 0.16 with mean 0.11 and for children extend between 0.04 and 0.25 with an average 0.17 (Table 4). While, the values of non-carcinogenic health risk of Cd in the drinking groundwater for adults range from 0.11 to 2.68 with an average of 1.29 and for children extend between 0.16 and 4.03 averaging 1.94 (Table 5). The Pb values for adults range from 0.04 to 0.20 with an average of 0.12 and for children vary from 0.07 to 0.31 averaging 0.18 (Table 5). According to USEPA (2011), Cd and Pb values of non-carcinogenic health risk should not exceed 1 to be considered as non-harmful drinking water.

The carcinogenic health risk values for Cd in the drinking surface water for adults range from $0.17*10^{-4}$ to $7.9*10^{-4}$ with an average of $3.9*10^{-4}$ and for children extend between $0.1*10^{-5}$ and $4.8*10^{-3}$ with mean $2.3*10^{-3}$ (Table 4). The Pb values for adults vary from $6.2*10^{-3}$ to $37*10^{-3}$ with mean $26*10^{-3}$ and for children range from $3.7*10^{-2}$ to $22*10^{-2}$ with an average $15*10^{-2}$ (Table 4). As well, 
[revised manuscript text omitted]

---

## Author Comment (AC4) · 22 Jan 2019

The response Dear editor, we are very grateful for those worthy comments, we would you like to publish our article in this valuable journal. We took these comments in our consideration and the response as the following: - 1) Indeed, the risk health of surface water samples not calculated, so that it is calculated and listed in Table No 4 in our article. 2) The hazard quotient (HQ) was mentioned in the material and method in page 5 lines 1 and 2 in equations and refers it in the health risk assessment in page 8 lines 7 and 8. 3) The current (Health Risk Assessment) work is a new approach in Egypt, didn't apply before. Both of Cd and Pb are derived from anthropogenic sources not

geogenic source in the study area. 4) This work is a part of Ph.D. thesis have been awarded in 2017, so it isn't up to date. For filtered and digested, we carried out this method according to APHA (1995) which is available then, we will do your advice about the digestion before filtration according to APHA (2017). The life time is the average of human being age is about 65 years for adults and 6.5 years for children. The reference of slope factor is USEPA (2011) is referred it in page 5 line 10. 5) Samples (No. S7 and No. S2) were collected from the canals, not the River Nile as listed in (Table 2). The levels of metals depend on the place of the sample where some samples were taken near mooring of boats which causes more pollution. Samples (No. S6 and No. S7) are collected from the River Nile (S6) and canal (S7) which not connected by the Sugar Factory outlet despite it is located near the factory. The sources of Cd and Pb are anthropogenic which is human activities such as drainage agricultures, sewage water, fuel stations, cesspits, garbage, fertilizers, pesticides mining and industrial waste which reached to water resources. Then, the source of Cd and Pb is derived from one source of anthropogenic. The majority of the study area is rural society lacks to the potable water network, so, the inhabitants depend on groundwater, canals and drains for drinking without any treatment. Groundwater recharges mainly from the River Nile, canals and drains so the level of Cd and Pb are homogenous in the surface and groundwater. We have been published two papers about evaluation both of surface water and groundwater for irrigation and industrial purposes. Source of maps is missed so that difficult increasing the resolution. We will attach the manuscript with the response. 6) Increasing references give scientific motivation for the article.

Last but not least we hope that our article to be published in your valuable journal.

Please also note the supplement to this comment:
https://www.drink-water-eng-sci-discuss.net/dwes-2018-37/dwes-2018-37-AC4-supplement.pdf

———————————————

37, 2018.

---

## Author Comment (AC5) · 22 Jan 2019

The response Dear editor and referee, we are very grateful for those worthy comments, we would you like to publish our article in this valuable journal. We took these comments in our consideration and the response as the following: - 1) No one apply the risk health on surface water and groundwater samples (HQ) in Egypt we are the first. If you want to prove that work a research. Indeed, many of researcher carried out studies on the study area but each one has a view that is the scientific research. Each editing in the text sentence has been done. Last but not least we hope that our article to be published in your valuable journal.

[Figure]

Please also note the supplement to this comment:
https://www.drink-water-eng-sci-discuss.net/dwes-2018-37/dwes-2018-37-AC5-
supplement.pdf

––––––––––––––––––––––––––––––––––
37, 2018.

[Figure]

**Supplement:**

[revised manuscript text omitted]
 are 0.38 and 0.055 mg/kg/day respectively. The average lifetime for an adult is 65 years and 6.5 years for children (USEPA, 2011).

**3. Results and Discussion**

**3.1. Surface water**

River Nile and its tributaries (canals and drains) are the main source of water in Egypt especially for the governorates allocated on the river banks and on its branches. Therefore, the quality of water was evaluated by measuring of Cd and Pb concentrations in the south of El-Minya Governorate during 2014 (Table 2). Cadmium concentrations (Table 2), ranged from 1 to 48µg/l, exceed the permissible limit (3µg/l) for drinking water according to WHO (2011). Excess Cd could accumulate in the kidney and remains for many years causes irreversible kidney damage (Goyer, 1996). Patients with kidney failure patients in the study area are presumed to have increased from 10 patient/million in 1974 to about 165 patient/million in 1995 and in 2005 it was 260 patient/million in El-Minya Governorate (El Minshawy and

Kamel, 2006). Agricultural activities are considered as the most important sources for Cd, where the Egyptian marine phosphorite used for the manufacture of super-phosphate fertilizers contains up to 20 ppm Cd (El-Kammar, 1974). Pesticides also can lead to high Cd content in the study area (Bowen, 1966).

[revised manuscript text omitted]

**3.3. Health risk assessment**

It was observed that general population in the rural area is using surface water from the rivers or canals and groundwater from hand pumps for drinking and domestic purposes because they don't have access to the tap water from the tube wells. Therefore, health risk assessment (HQ) for surface water and groundwater was carried out in this study. The results of non-carcinogenic and carcinogenic health risks (HQ) due to metal exposure in surface water and groundwater samples are provided in Tables (4) and (5).

The non-carcinogenic health risk values for Cd in the drinking surface water for adults vary from 0.05 to 2.06 with an average of 1.30 and for children fluctuate between 0.08 and 3.95 with mean 1.96 (Table 4). The Pb values for adults range from 0.03 to 0.16 with mean 0.11 and for children extend between 0.04 and 0.25 with an average of 0.17 (Table 4). While, the values of the non-carcinogenic health risk of Cd in the drinking groundwater for adults range from 0.11 to 2.68 with an average of 1.29 and for children extend between 0.16 and 4.03 averaging 1.94 (Table 5). The Pb values for adults range from 0.04 to 0.20 with an average of 0.12 and for children vary from 0.07 to 0.31 averaging 0.18 (Table 5). According to USEPA (2011), Cd and Pb values of non-carcinogenic health risk should not exceed 1 to be considered as non-harmful drinking water.

The carcinogenic health risk values for Cd in the drinking surface water for adults range from $0.17*10^{-4}$ to $7.9*10^{-4}$ with an average of $3.9*10^{-4}$ and for children extend between $0.1*10^{-5}$ and $4.8*10^{-3}$ with mean $2.3*10^{-3}$ (Table 4). The Pb values for adults vary from

6.2*10$^{-3}$ to 37*10$^{-3}$ with mean 26*10$^{-3}$ and for children range from 3.7*10$^{-2}$ to 22*10$^{-2}$ with an average 15*10$^{-2}$ (Table 4). As well, 
[revised manuscript text omitted]

[Figure]

2          **Figure 1: Location map of the study area.**

[Figure]

Figure 2: Geologic map of the study area.

[Figure]

Figure 3: Spatial distribution map of Cd in the studied groundwater samples.

[Figure]

**Figure 4: Spatial distribution map of Pb in the studied groundwater samples.**

**Table 1: Surface water samples and localities.**

| Sample No. | Canals | Sample No. | Canals & Drains |
|---|---|---|---|
| $S_3$, $S_6$, $S_{30}$ | River Nile | $S_{28}$ | Al Nasriyah canal |
| $S_1$, $S_5$ | Al Ibrahimayah canals | $S_9$ | Branch from Ismail Siri canal |
| $S_{11}$, $S_{20}$, $S_{24}$ | Bahr Youssef | $S_{27}$ | Branch from Al Nasriyah canal |
| $S_2$ | Al Sawahliyah canals | $S_4$ | Al Sellic Drain |
| $S_7$ | Hafez canal | $S_{14}$ | Jiris Drain |
| $S_8$ | Al ShiekhSharf canal | $S_{15}$ | Al Moheet Drain |
| $S_{10}$, $S_{16}$, $S_{19}$ | Al Ashmoneen canal | $S_{18}$, $S_{29}$ | Kabkab Drain |
| $S_{12}$, $S_{17}$ | Ismail Siri canal | $S_{21}$ | Al umomi Drain |
| $S_{13}$ | Al Mantutiyah canal | $S_{22}$ | Al Sangrawi Drain |
| $S_{23}$ | Al Badraman canal | $S_{26}$ | Tandah Drain |
| $S_{25}$ | Al Arosiyah canal | | |

**Table 2:  Cd and Pb concentrations in the surface water samples (µg/l).**

| Sample No. | Nile & Canals | | Drains | | Sample No. | Nile & Canals | | Drains | |
|---|---|---|---|---|---|---|---|---|---|
| | Cd | Pb | Cd | Pb | | Cd | Pb | Cd | Pb |
| $S_1$ | 39 | 184 | - | - | $S_{18}$ | - | - | 17 | 216 |
| $S_2$ | 37 | 210 | - | - | $S_{19}$ | 15 | 208 | - | - |
| $S_3$ | 22 | 270 | - | - | $S_{20}$ | 15 | 208 | - | - |
| $S_4$ | - | - | 22 | 222 | $S_{21}$ | - | - | 28 | 183 |
| $S_5$ | 19 | 241 | - | - | $S_{22}$ | - | - | 21 | 260 |
| $S_6$ | 1 | 54 | - | - | $S_{23}$ | 33 | 275 | - | - |
| $S_7$ | 19 | 96 | - | - | $S_{24}$ | 21 | 250 | - | - |
| $S_8$ | 48 | 288 | - | - | $S_{25}$ | 7 | 272 | - | - |
| $S_9$ | 9 | 209 | - | - | $S_{26}$ | - | - | 22 | 273 |
| $S_{10}$ | 20 | 222 | - | - | $S_{27}$ | 16 | 298 | - | - |
| $S_{11}$ | 19 | 268 | - | - | $S_{28}$ | 35 | 329 | - | - |
| $S_{12}$ | 43 | 262 | - | - | $S_{29}$ | - | - | 23 | 198 |
| $S_{13}$ | 28 | 303 | - | - | $S_{30}$ | 28 | 192 | - | - |
| $S_{14}$ | - | - | 42 | 234 | Mean | 24 | 233 | 23 | 224 |
| $S_{15}$ | - | - | 12 | 213 | Median | 21 | 245 | 22 | 219 |
| $S_{16}$ | 32 | 225 | - | - | Min. | 1 | 54 | 12 | 183 |
| $S_{17}$ | 22 | 261 | - | - | Max. | 48 | 329 | 42 | 273 |

**Table 3: Cd and Pb concentrations (µg/l) in the study groundwater samples.**

| Sample No. | Cd | Pb | Sample No. | Cd | Pb |
|---|---|---|---|---|---|
| $G_1$ | 15 | 249 | $G_{15}$ | 24 | 274 |
| $G_2$ | 2 | 90 | $G_{16}$ | 8 | 242 |
| $G_3$ | 19 | 410 | $G_{17}$ | 19 | 228 |
| $G_4$ | 20 | 224 | $G_{18}$ | 19 | 228 |
| $G_5$ | 26 | 252 | $G_{19}$ | 19 | 243 |
| $G_6$ | 23 | 242 | $G_{20}$ | 28 | 162 |
| $G_7$ | 36 | 296 | $G_{21}$ | 14 | 266 |
| $G_8$ | 25 | 230 | $G_{22}$ | 27 | 193 |
| $G_9$ | 39 | 213 | $G_{23}$ | 17 | 275 |
| $G_{10}$ | 29 | 229 | $G_{24}$ | 19 | 264 |
| $G_{11}$ | 49 | 254 | $G_{25}$ | 42 | 276 |
| $G_{12}$ | 9 | 197 | Mean | 24 | 242 |
| $G_{13}$ | 38 | 270 | Minimum | 2 | 90 |
| $G_{14}$ | 23 | 242 | Maximum | 49 | 410 |

**Table 4: Statistical parameters of non-carcinogenic and carcinogenic health risks for surface water samples.**

| Parameter | Non-carcinogenic for Adults | | Non-carcinogenic for Children | |
|---|---|---|---|---|
| | HQ Cd | HQ Pb | HQ Cd | HQ Pb |
| **Minimum** | 0.05 | 0.03 | 0.08 | 0.04 |
| **Maximum** | 2.06 | 0.16 | 3.95 | 0.25 |
| **Average** | 1.30 | 0.11 | 1.96 | 0.17 |
| **Parameter** | Carcinogenic for Adults | | Carcinogenic for Children | |
| | HQ Cd | HQ Pb | HQ Cd | HQ Pb |
| **Minimum** | $0.17 \times 10^{-4}$ | $6.2 \times 10^{-3}$ | $0.1 \times 10^{-5}$ | $3.7 \times 10^{-2}$ |
| **Maximum** | $7.9 \times 10^{-4}$ | $37 \times 10^{-3}$ | $4.8 \times 10^{-3}$ | $22 \times 10^{-2}$ |
| **Average** | $3.9 \times 10^{-4}$ | $26 \times 10^{-3}$ | $2.3 \times 10^{-3}$ | $15 \times 10^{-2}$ |

HQ = Hazard quotient;     $HQ_{\text{non-carcinogenic}} = CDI / RfD$;     $HQ_{\text{carcinogenic}} = CDI / SF$

**Table 5: Statistical parameters of non-carcinogenic and carcinogenic health risks for groundwater samples.**

| Parameter | Non-carcinogenic for Adults | | Non-carcinogenic for Children | |
|-----------|-------|-------|-------|-------|
| | HQ Cd | HQ Pb | HQ Cd | HQ Pb |
| Minimum | 0.11 | 0.04 | 0.16 | 0.07 |
| Maximum | 2.68 | 0.20 | 4.03 | 0.31 |
| Average | 1.29 | 0.12 | 1.94 | 0.18 |
| **Parameter** | **Carcinogenic for Adults** | | **Carcinogenic for Children** | |
| | HQ Cd | HQ Pb | HQ Cd | HQ Pb |
| Minimum | $4.8*10^{-6}$ | $3.13*10^{-5}$ | $2.9*10^{-5}$ | $1.67*10^{-9}$ |
| Maximum | $1.2*10^{-4}$ | $14.3*10^{-5}$ | $7.11*10^{-4}$ | $86.1*10^{-5}$ |
| Average | $5.66*10^{-5}$ | $8.41*10^{-5}$ | $3.42*10^{-4}$ | $48.5*10^{-5}$ |

HQ = Hazard quotient;     $HQ_{\text{non-carcinogenic}} = CDI / RfD$;     $HQ_{\text{carcinogenic}} = CDI / SF$

---

## Author Comment (AC6) · 27 Jan 2019

Dear Reviewer Thank you very much for your efforts in reviewing this work and for your valuable comments which assist us to improve our article quality 1-In this paper Cd and Pb are monitored in groundwater and surface water and compared to WHO guidelines. The paper is poorly written and not very innovative, since most of the sampling was already done before. Through this work we applied human health risk assessment which nearly used for the first time in Egypt. I has great advantages than comparison with WHO guidelines. For example Pb exceeded WHO guidelines but according to the calculated HQnc it hasn't any health problem 2-General comments: -

Give clear objective (and knowledge gap) at the end of the introduction. The objectives were rewritten at the abstract and the end of introduction 3- "Location" should be part of the Materials and Methods section (not of introduction) We moved it into the Materials and Methods section 4- Explain why Health Risk Assessment gives other information than WHO guidelines Because, it takes in consideration; age, body weight, metal concentration, exposure duration and daily intake. Page 2 lines20-21 5- Check language, including tenses. - Introduce abbreviations (like Cd and Pb) once and then use the abbreviations in the rest of the manuscript. - Structure the description of Cd and Pb in the introduction in the same way. - Use same structure for describing Cd and Pb (as on pg 2) - Avoid repetitions in the paper (like first 4 sentences on pg 3). All these comments were taken in consideration and done 6-Specific comments: - Pg 1, line 17, delete "in addition,: : : human activities" from abstract. - Pg1, line 21, 2x "pollutions" - Pg 1, line 22, amount = concentrations - Pg 1, line 24, content = concentrations - Pg 2, line 19, delete second "body" - Pg 5, line 16-17, delete sentence "the close: : :. their source". - Pg 5, line 19-20, not relevant - Pg 5, line 24, insert "important" between "most" and "source" - Pg 6, line 4, "throwing" = "deposition" - Pg 6, line 5, "picked" = "collected" - Pg 6, line 13-15, not relevant here. - Pg 6, line 24, "rubbish" = "solid waste". - Pg 7, line 2, "second water source": : : "and the only one in the desert fringes" - Pg 7, line 3, delete "unfortunately" - Pg 7, line 4, delete "which lacking safe potable water source". - Pg 7, line 7, "absorbance" = "adsorption" - Pg 7, line 10, ": : :desert road, is vulnerable as a result of the unconfined condition of the aquifers". - Pg 7, line 14, which type of anthropogenic activities? - Pg 7, line 24, seem to be very high values.. - Pg 9, therefore treatment is needed before water containing Cd and Pb can be used for drinking. All these comments were taken in consideration and done

Please also note the supplement to this comment:
https://www.drink-water-eng-sci-discuss.net/dwes-2018-37/dwes-2018-37-AC6-supplement.pdf

[Figure]

[Figure]

**Supplement:**

**Evaluation of human risks of surface water and groundwater contaminated with Cd and Pb south of El-Minya Governorate, Egypt.**

Salman A. Salman[1], Ahmed A. Asmoay[1], Amr El-Gohary[1], Hassan Sabet[2]

[1] Geological Sciences Dep. National Research Centre, Dokki, Giza, 12622, Egypt.

[2] Geology Dep., Faculty of Science, Al-Azhar University, Cairo, Egypt.

*Correspondence to*: Ahmed A. Asmoay (asmoay@gmail.com)

**Abstract:**

Water pollution with cadmium (Cd) and lead (Pb) has worldwide concern because of their health impact. Determination of their concentrations and potential human health risks in surface and groundwater at south El-Minya Governorate, Egypt is the main aim of this study. Fifty-five samples were collected; 30 samples surface water and 25 samples groundwater. The samples were analyzed using Atomic Absorption Spectrometry (AAS) to determine Cd and Pb contents. Their levels in surface and groundwater exceeded the maximum allowable level for drinking water which set by WHO. The hazard quotient (HQ) showed that the surface and groundwater may be pose health risk to residents, especially the children. However, the water can be used safely for irrigation.

**1. Introduction**

Water resources pollution was becoming a worldwide problem. To protect the environment and public health, it is important to have precise knowledge on concentration and type of water pollutants especially heavy metals. Because the heavy metals have long biological half-life, they are threat the human health in case of excessive concentrations (Albji et al., 2013). Cd and Pb are of the most chemical pollutants that threaten the water quality for different uses. Monitoring of their concentrations in water is of critical importance for

protecting ecological and human health, because of their harmful effects and persistence (Nazar, et. al., 2012).

Cadmium has a major environmental concern and ranked as the sixth significant human health hazard toxic substances (ATSDR, 1997). It is released into the aqueous system from metal plating, smelting, mining, cadmium-nickel batteries, phosphate fertilizers, paint industries, pigments and alloy industries as well as from sewage (Kadirvalu and Namasivayam, 2003). The nervous system appears to be the most sensitive target of Cd toxicity. Cadmium exposure can produce a wide variety of acute and chronic effects in humans such as renal failure, lung insufficiency, bone lesions and hypertension (Sun and Li, 2011).

Pb is the most common environmental contaminant (Chiang et. al, 2012). It doesn't undergo degradation or decomposition. Thus, its long persistence in the environment exacerbates its threat to human health. Pb is used in many industries including lead smelting and processing, batteries manufacture, pigments, solder, plastics, cable sheathing, fuel, ammunition and ceramics. Due to urbanization, Pb and other metals are regularly discharge into fields, water and soils through sewage sludge, urban runoff and automobile exhaust (Elnazer et al. 2015). Pb absorbed by human body disturbs many processes and is harmful to many organs such as heart, bones and nervous systems (Bruce et. al., 2012).

The application of human health risk assessment model has many advantages than comparison with drinking water guidelines as WHO guideline. Because, it takes in consideration the age, body weight, metal concentration, exposure duration and daily intake. So, quantifying human risk (carcinogenic and non-carcinogenic risks) of Cd and Pb to children and adults is important issue. The hazard quotient (HQ) of the USEPA (1989) is extensively used to characterize the carcinogenic and non-carcinogenic health effects of toxic metals by comparison of their exposure effects to a reference dose (R$f$D) (Qu et al., 2012). It

was documented in many investigations by taking into consideration exposure scenarios of metal intake through contaminated water (Muhammad et al., 2011; Shah et al., 2012; Dou and Li, 2012). So, the objective of the present study is the determination of Cd and Pb concentrations and potential human health risks in the surface and the ground water systems in the western part of the River Nile between Abu Qurqas and Dyer Mawas districts, south El-Minya Governorate, Egypt.

**2. Material and methods**

**2.1. Location**

The study area occupied the middle part of the Nile Valley between longitudes 30° 29′ and 30° 54′E and latitudes 27° 37′and 27° 56′N (Fig. 1). It is bounded by the River Nile from the east and the calcareous plateau at the west between Abu Qurqas northward and Dyer Mawas at the south. The water resources in the study area are represented by the River Nile, canals and drains as well as groundwater (Fig. 1). The stratigraphic succession (Fig. 2) in El-Minya area is represented by Tertiary and Quaternary sedimentary rocks (Said 1981). The main aquifer in the study area is represented by Pleistocene sediments which compose of sand and gravel of different sizes with some clay intercalation (Sadek 2001). The aquifer is semi-confined in the old cultivated land (at the eastern part of the study area) and unconfined in the desert fringes (at the western part of the study area). The groundwater flows generally from the southern part to the northern part of the study area. The aquifer is recharged by Nile water, irrigation system, drains, agricultural infiltration and vertical upward from the deeper saline aquifers (Abdalla et al. 2009). The main sources of water for different purposes in the study area are Nile, canals, drains and groundwater. The study area contains many industrial zones, agricultural activities and urban areas.

**2.2. Sampling and analyses**

In November 2014, thirty water samples were collected from surface water resources at the study area (Fig. 1 and Table. 1). In addition, 25 groundwater samples were collected from the Quaternary aquifer (Fig. 1). Pre-rinsed polypropylene bottles were filled with the samples sealed tightly, acidified (pH < 2) with nitric acid to prevent precipitation, microbial activity and sorption losses to container walls.  At lab the samples were filtered through filter paper (Whatman No. 42) and digested with nitric acid (APHA, 1995). Samples were analyzed using atomic absorption spectrometer instrument (model: Perkin Elmer 400) in National Research Centre Laboratories.

For health risk assessment, non-carcinogenic ($HQ_{nc}$) and carcinogenic ($HQ_c$) hazard quotient for each contaminant was calculated according to the following equations (Kelepertzis 2014):

$$HQ_{nc}= CDI / RfD \tag{1}$$

$$HQ_c = CDI *SF \tag{2}$$

$$CDI = C*IR*ED*EF/BW*AT \tag{3}$$

[revised manuscript text omitted]

**3.3. Health risk assessment**

Water from the River Nile and canals as well as groundwater is used for drinking and domestic purposes. Unfortunately, the applied treatment techniques (flocculation and coagulation with alum) in drinking water stations (Donia 2007) in the study area are ineffective for the removal of toxic metals (Fatoki and Ogunfowokan 2002). It was observed, too, that great number of residents in rural areas use groundwater from hand pumps for drinking because they don't have access to the tap water. El-Minshawy and Kamel (2006) mentioned that the use of unsafe water for drinking contributes up to 71.8% of the renal failure in the study area. Therefore, health risk assessment surface water and groundwater was carried out in this study. The results of $HQ_{nc}$ and $HQ_c$ due to metal exposure in surface water and groundwater samples are provided in Tables (4) and (5).

The calculated $HQ_{nc}$ average values for Cd in surface water for adults and children were 1.30 and 1.96, respectively while; $HQ_{nc}$ average values for Pb were 0.11 and 0.17 (Table 4). On the other hand, the calculated $HQ_{nc}$ average values for Cd in groundwater for adults and children were 1.29 and 1.94, respectively while; $HQ_{nc}$ average values for Pb were 0.12 and 0.18 (Table 5). According to USEPA (2011), Cd can cause non-carcinogenic health

problems because its $HQ_{nc}$ values were more than 1. While Pb presence in water hasn't adverse health impacts ($HQ_{nc}<1$), however its concentration exceeded WHO (2011) guidelines.

The calculated $HQ_c$ values for Cd in the surface water for adults range from $0.17*10^{-4}$ to $7.9*10^{-4}$ and for children from $0.1*10^{-5}$ to $4.8*10^{-3}$ (Table 4). The calculated $HQ_c$ values for Pb varied from $6.2*10^{-3}$ to $37*10^{-3}$ for adults and varied from $3.7*10^{-2}$ to $22*10^{-2}$ for children (Table 4). While, the calculated $HQ_c$ values for Cd in the groundwater ranged from $4.8*10^{-6}$ to $1.2*10^{-4}$ for adults and from $2.9*10^{-5}$ to $7.11*10^{-4}$ for children (Table 5). the calculated $HQ_c$ values for Pb varied between $3.13*10^{-5}$ and $14.3*10^{-5}$ for adults and varied from $1.67*10^{-9}$ to $86.1*10^{-5}$ for children (Tables 5). According to USEPA (2011) recommenden value ($HQ_c<10^{-6}$) the studied surface and groundwater consumption could cause carcinogenic health risks in adults and children.

**4. Conclusion**

Cadmium and lead contents of the studied samples from River Nile, canals and drains as well as groundwater exceed the permissible limits for drinking water and could be disturbing many adverse health impacts. The calculated $HQ_{nc}$ indicted the health hazards of the presence of Cd in water resources in the study area in opposite to the presence of Pb. This result supports the importance of application of health risk assessment model than the comparison with drinking water guidelines. Unfortunately, the presence of Cd and Pb in surface and groundwater of the study area have carcinogenic health impacts. The water resources in the study area (surface and groundwater) are suitable for irrigation purposes. Source of pollution in the investigated area were derived from anthropogenic activities such as industries, agriculture, mining and urban runoff.

It is recommended to apply effective treatment agents for removing toxic metals in drinking water stations, connect the houses in rural parts of the study area with safe drinking water lines and regular monitoring of water resources and the end user water lines.

**Acknowledgments**

The authors gratefully acknowledge the National Research Centre for funding this research as a Ph.D. internal project and the grant no. is (8/5/9) to support Mr. Ahmed A. Asmoay to do the lab work.

[revised manuscript text omitted]

1     **Table 5: Statistical parameters of non-carcinogenic and carcinogenic health risks for**
2     **groundwater samples.**

| Parameter | $HQ_{nc}$ for Adults | | $HQ_{nc}$ for Children | |
|---|---|---|---|---|
| | Cd | Pb | Cd | Pb |
| Minimum | 0.11 | 0.04 | 0.16 | 0.07 |
| Maximum | 2.68 | 0.20 | 4.03 | 0.31 |
| Average | 1.29 | 0.12 | 1.94 | 0.18 |
| Parameter | $HQ_c$ for Adults | | $HQ_c$ for Children | |
| | Cd | Pb | Cd | Pb |
| Minimum | $4.8*10^{-6}$ | $3.13*10^{-5}$ | $2.9*10^{-5}$ | $1.67*10^{-9}$ |
| Maximum | $1.2*10^{-4}$ | $14.3*10^{-5}$ | $7.11*10^{-4}$ | $86.1*10^{-5}$ |
| Average | $5.66*10^{-5}$ | $8.41*10^{-5}$ | $3.42*10^{-4}$ | $48.5*10^{-5}$ |

---

## Author Comment (AC7) · 27 Jan 2019

Dear Reviewer, We are very grateful for those worthy comments. We took these comments in our consideration and the response as the following:- 1) The value of this risk is not emphasized, as it was calculated only for the groundwater that be used with unidentified inhabitants and the treatment of the surface water did not put into account. Risk has calculated for surface water and listed in Table No 4 in our article and discussed in pages 7&8. Unfortunately, the applied treatment techniques (flocculation and coagulation with alum) in drinking water stations (Donia 2007) in the study area are ineffective for the removal of toxic metals (Fatoki and Ogunfowokan 2002). 2) In

the Abstract, authors referred to the hazard index (the summation of HQ) (Page 1, Line 15), while there is no information in the whole manuscript about this parameter and how they calculate it. The hazard quotient (HQ) was mentioned in the material and method in page 4 lines 10-16 and calculated using equations 1&2. 3) In the Introduction: # (Page 3, Line 10), the aim should be the same as written in the Abstract. Also, authors did not show the novelty of the work. What is done before in risk assessment research work? # (Page 3, Line 22 until the end of the paragraph) the description of the nature of rocks and stones in the area did not connected with the presence of the Cd and Pb problem. -The aim was rewritten as the abstract. This work (health risk assessment) is a new in Egypt didn't apply before, the previous works dealt with comparison with national and international drinking water guidelines. -The description of rocks was deleted and only short description of aquifer and geology was written. 4) Materials and Methods: # (Page 4, Line 14) the samples were taken in November 2014. The results must be up-to-date, unless the authors mention that there is no action or changing for the situation yet. # Authors mentioned that they filtered the collected samples and then digested them with nitric acid. This is give the concentration of heavy metals in the water filtrate (dissolved heavy metals). The samples should acidified first or digested and then filtered to have the total and exact heavy metal concentration. Also, the standard method that they refer for this analysis is not up-to-date. APHA, 2017 is recommended. # (Page 5, Line 5), the exposure frequency is 350 day/year. # (Page 5, Line 6), how much lifetime is used for the calculations? # (Equation 3), the slope factor (SF) is not defined and the reference for SF values used for Cd and Pb is missed. -This work is part of PhD thesis has been awarded in 2017, and the problem is still now. - Samples acidified (pH < 2) with nitric acid to prevent precipitation, microbial activity and sorption losses to container walls. Page 4, line 5. - APHA (1995) book is the available at our lab. -The life time is the average of human age is about 65 year for adults and 6.5 year for children. Page 4 lines 24-25 -The reference of slope factor is USEPA (2011) is referred it in page 4 line 23. 5) Results and Discussion # (Page 6, Line 5), Why the Cd concentrations in the River Nile (S6, S7) are lower than that for River Nile (S2,

S3)?. While S2 and S3 are from the southern part and even S6 and S7 sites are near to Abu Qurqas Sugar Factory. Authors claim that this Sugar Factory is a source of Cd pollutants. # Authors did not explain the source of Pb in the River Nile, especially at S2 and S3 sites. # S4 appears in the map (Fig. 1) as a canal site and written in the Table 2 as a drain site. # Authors refer in Page 5, Line 17 to the unity of Cd and Pb source and they mentioned in the same page 5, Line 23 that the agricultural activities (Fertilizers and pesticides) are the main sources for Cd. Then they mentioned in Page 6 Line 21 that the Cd and Pb concentrations in the River Nile and canals are higher than that in the drains. They explained that due to the high human activities. Actually, what is the main source of Cd and Pb pollution? # What is the source of Pb in the River Nile samples (S2&S3)? # The area under study is covered by drinking water distribution network. There is no information about that. In Page 7, Line 4, how many inhabitants not covered by the DW network and safe potable water? And how they are far from the distribution system? # Why the levels of Cd and Pb in the groundwater sites near to the Western Deseret Road are close to that in River Nile while these wells are far from the River by about 25 Km and far from the canal (Bahr El Youssif) by about 15 Km. # The health risk assessment in this research article is based only on the exposure of some inhabitants in some villages on the study area with Cd and Pb via ingestion of contaminated groundwater. The study ignored the contact with the surface water in irrigation activities and did not give full picture for the potable water resources, especially that there are water treatment facilities covering the area. # The resolution of maps is so low -Sample No S2 and S7 were collected from the canals and samples No S3 and S6 are collected from the River Nile stream as listed in Table 2. The levels of metals depend on the place of sample where some samples taken near navigation sites and residential areas which cause more pollution. (S6) and canal (S7) which not connected by the Sugar Factory outlet despite it is located near the Factory. -S4 (Al Sellic Drain) is very close to Al Ibrahimayah canals (about 50 meters), so this drain not appears on the map. -The authors mean by unity that these metals come from anthropogenic activities. These activities include the application of agrochemicals (fertilizers, herbicides and pesticides), vehicle exhaust, urban runoff and industrial activities. - The study area covered by drinking water distribution network but some rural areas which contain thousands of residents aren't on this network. Also, the applied treatment techniques (flocculation and coagulation with alum) in drinking water stations (Donia 2007) in the study area are ineffective for the removal of toxic metals (Fatoki and Ogunfowokan 2002). In addition the desert area doesn't have drinking water network. Page 7 lines 14-16 -The source of Pb in the River Nile samples (S3) and Al Sawahliyah canal (S2) is urban runoff and vehicle exhaust. -The level of Cd and Pb are nearly homogenous in groundwater of the study area except some hot spots as a result of uncoffining and great vulnerability of aquifer in the desert area and great polluting activities in the old cultivated land. - This work dealt with the use of water for drinking and we take into consideration the surface water risk. - The resolution of maps enhanced. We made low resolution for upload problem. 6) The number of references (40) is so high for research article. The increased number of references was a result of my discussion with my colleagues in my lab to support my interpretation but your advice to reduce the references was taken in consideration (references reduced to 32 references).

.

Please also note the supplement to this comment:
https://www.drink-water-eng-sci-discuss.net/dwes-2018-37/dwes-2018-37-AC7-supplement.pdf

**Supplement:**

**Evaluation of human risks of surface water and groundwater contaminated with Cd and Pb south of El-Minya Governorate, Egypt.**

Salman A. Salman[1], Ahmed A. Asmoay[1], Amr El-Gohary[1], Hassan Sabet[2]

[1] Geological Sciences Dep. National Research Centre, Dokki, Giza, 12622, Egypt.

[2] Geology Dep., Faculty of Science, Al-Azhar University, Cairo, Egypt.

*Correspondence to*: Ahmed A. Asmoay (asmoay@gmail.com)

**Abstract:**

Water pollution with cadmium (Cd) and lead (Pb) has worldwide concern because of their health impact. Determination of their concentrations and potential human health risks in surface and groundwater at south El-Minya Governorate, Egypt is the main aim of this study. Fifty-five samples were collected; 30 samples surface water and 25 samples groundwater. The samples were analyzed using Atomic Absorption Spectrometry (AAS) to determine Cd and Pb contents. Their levels in surface and groundwater exceeded the maximum allowable level for drinking water which set by WHO. The hazard quotient (HQ) showed that the surface and groundwater may be pose health risk to residents, especially the children. However, the water can be used safely for irrigation.

**1. Introduction**

Water resources pollution was becoming a worldwide problem. To protect the environment and public health, it is important to have precise knowledge on concentration and type of water pollutants especially heavy metals. Because the heavy metals have long biological half-life, they are threat the human health in case of excessive concentrations (Albji et al., 2013). Cd and Pb are of the most chemical pollutants that threaten the water quality for different uses. Monitoring of their concentrations in water is of critical importance for

protecting ecological and human health, because of their harmful effects and persistence (Nazar, et. al., 2012).

Cadmium has a major environmental concern and ranked as the sixth significant human health hazard toxic substances (ATSDR, 1997). It is released into the aqueous system from metal plating, smelting, mining, cadmium-nickel batteries, phosphate fertilizers, paint industries, pigments and alloy industries as well as from sewage (Kadirvalu and Namasivayam, 2003). The nervous system appears to be the most sensitive target of Cd toxicity. Cadmium exposure can produce a wide variety of acute and chronic effects in humans such as renal failure, lung insufficiency, bone lesions and hypertension (Sun and Li, 2011).

Pb is the most common environmental contaminant (Chiang et. al, 2012). It doesn't undergo degradation or decomposition. Thus, its long persistence in the environment exacerbates its threat to human health. Pb is used in many industries including lead smelting and processing, batteries manufacture, pigments, solder, plastics, cable sheathing, fuel, ammunition and ceramics. Due to urbanization, Pb and other metals are regularly discharge into fields, water and soils through sewage sludge, urban runoff and automobile exhaust (Elnazer et al. 2015). Pb absorbed by human body disturbs many processes and is harmful to many organs such as heart, bones and nervous systems (Bruce et. al., 2012).

The application of human health risk assessment model has many advantages than comparison with drinking water guidelines as WHO guideline. Because, it takes in consideration the age, body weight, metal concentration, exposure duration and daily intake. So, quantifying human risk (carcinogenic and non-carcinogenic risks) of Cd and Pb to children and adults is important issue. The hazard quotient (HQ) of the USEPA (1989) is extensively used to characterize the carcinogenic and non-carcinogenic health effects of toxic metals by comparison of their exposure effects to a reference dose (R$f$D) (Qu et al., 2012). It

was documented in many investigations by taking into consideration exposure scenarios of metal intake through contaminated water (Muhammad et al., 2011; Shah et al., 2012; Dou and Li, 2012). So, the objective of the present study is the determination of Cd and Pb concentrations and potential human health risks in the surface and the ground water systems in the western part of the River Nile between Abu Qurqas and Dyer Mawas districts, south El-Minya Governorate, Egypt.

**2. Material and methods**

**2.1. Location**

The study area occupied the middle part of the Nile Valley between longitudes 30° 29′ and 30° 54′E and latitudes 27° 37′and 27° 56′N (Fig. 1). It is bounded by the River Nile from the east and the calcareous plateau at the west between Abu Qurqas northward and Dyer Mawas at the south. The water resources in the study area are represented by the River Nile, canals and drains as well as groundwater (Fig. 1). The stratigraphic succession (Fig. 2) in El-Minya area is represented by Tertiary and Quaternary sedimentary rocks (Said 1981). The main aquifer in the study area is represented by Pleistocene sediments which compose of sand and gravel of different sizes with some clay intercalation (Sadek 2001). The aquifer is semi-confined in the old cultivated land (at the eastern part of the study area) and unconfined in the desert fringes (at the western part of the study area). The groundwater flows generally from the southern part to the northern part of the study area. The aquifer is recharged by Nile water, irrigation system, drains, agricultural infiltration and vertical upward from the deeper saline aquifers (Abdalla et al. 2009). The main sources of water for different purposes in the study area are Nile, canals, drains and groundwater. The study area contains many industrial zones, agricultural activities and urban areas.

**2.2. Sampling and analyses**

In November 2014, thirty water samples were collected from surface water resources at the study area (Fig. 1 and Table. 1). In addition, 25 groundwater samples were collected from the Quaternary aquifer (Fig. 1). Pre-rinsed polypropylene bottles were filled with the samples sealed tightly, acidified (pH < 2) with nitric acid to prevent precipitation, microbial activity and sorption losses to container walls.  At lab the samples were filtered through filter paper (Whatman No. 42) and digested with nitric acid (APHA, 1995). Samples were analyzed using atomic absorption spectrometer instrument (model: Perkin Elmer 400) in National Research Centre Laboratories.

For health risk assessment, non-carcinogenic ($HQ_{nc}$) and carcinogenic ($HQ_c$) hazard quotient for each contaminant was calculated according to the following equations (Kelepertzis 2014):

$$HQ_{nc}= CDI / RfD \tag{1}$$

$$HQ_c = CDI *SF \tag{2}$$

$$CDI = C*IR*ED*EF/BW*AT \tag{3}$$

[revised manuscript text omitted]

**3.3. Health risk assessment**

Water from the River Nile and canals as well as groundwater is used for drinking and domestic purposes. Unfortunately, the applied treatment techniques (flocculation and coagulation with alum) in drinking water stations (Donia 2007) in the study area are ineffective for the removal of toxic metals (Fatoki and Ogunfowokan 2002). It was observed, too, that great number of residents in rural areas use groundwater from hand pumps for drinking because they don't have access to the tap water. El-Minshawy and Kamel (2006) mentioned that the use of unsafe water for drinking contributes up to 71.8% of the renal failure in the study area. Therefore, health risk assessment surface water and groundwater was carried out in this study. The results of $HQ_{nc}$ and $HQ_c$ due to metal exposure in surface water and groundwater samples are provided in Tables (4) and (5).

The calculated $HQ_{nc}$ average values for Cd in surface water for adults and children were 1.30 and 1.96, respectively while; $HQ_{nc}$ average values for Pb were 0.11 and 0.17 (Table 4). On the other hand, the calculated $HQ_{nc}$ average values for Cd in groundwater for adults and children were 1.29 and 1.94, respectively while; $HQ_{nc}$ average values for Pb were 0.12 and 0.18 (Table 5). According to USEPA (2011), Cd can cause non-carcinogenic health

problems because its $HQ_{nc}$ values were more than 1. While Pb presence in water hasn't adverse health impacts ($HQ_{nc}<1$), however its concentration exceeded WHO (2011) guidelines.

The calculated $HQ_c$ values for Cd in the surface water for adults range from $0.17*10^{-4}$ to $7.9*10^{-4}$ and for children from $0.1*10^{-5}$ to $4.8*10^{-3}$ (Table 4). The calculated $HQ_c$ values for Pb varied from $6.2*10^{-3}$ to $37*10^{-3}$ for adults and varied from $3.7*10^{-2}$ to $22*10^{-2}$ for children (Table 4). While, the calculated $HQ_c$ values for Cd in the groundwater ranged from $4.8*10^{-6}$ to $1.2*10^{-4}$ for adults and from $2.9*10^{-5}$ to $7.11*10^{-4}$ for children (Table 5). the calculated $HQ_c$ values for Pb varied between $3.13*10^{-5}$ and $14.3*10^{-5}$ for adults and varied from $1.67*10^{-9}$ to $86.1*10^{-5}$ for children (Tables 5). According to USEPA (2011) recommenden value ($HQ_c<10^{-6}$) the studied surface and groundwater consumption could cause carcinogenic health risks in adults and children.

**4. Conclusion**

Cadmium and lead contents of the studied samples from River Nile, canals and drains as well as groundwater exceed the permissible limits for drinking water and could be disturbing many adverse health impacts. The calculated $HQ_{nc}$ indicted the health hazards of the presence of Cd in water resources in the study area in opposite to the presence of Pb. This result supports the importance of application of health risk assessment model than the comparison with drinking water guidelines. Unfortunately, the presence of Cd and Pb in surface and groundwater of the study area have carcinogenic health impacts. The water resources in the study area (surface and groundwater) are suitable for irrigation purposes. Source of pollution in the investigated area were derived from anthropogenic activities such as industries, agriculture, mining and urban runoff.

It is recommended to apply effective treatment agents for removing toxic metals in drinking water stations, connect the houses in rural parts of the study area with safe drinking water lines and regular monitoring of water resources and the end user water lines.

**Acknowledgments**

The authors gratefully acknowledge the National Research Centre for funding this research as a Ph.D. internal project and the grant no. is (8/5/9) to support Mr. Ahmed A. Asmoay to do the lab work.

[revised manuscript text omitted]

1     **Table 5: Statistical parameters of non-carcinogenic and carcinogenic health risks for**
2     **groundwater samples.**

| Parameter | $HQ_{nc}$ for Adults | | $HQ_{nc}$ for Children | |
|---|---|---|---|---|
| | Cd | Pb | Cd | Pb |
| Minimum | 0.11 | 0.04 | 0.16 | 0.07 |
| Maximum | 2.68 | 0.20 | 4.03 | 0.31 |
| Average | 1.29 | 0.12 | 1.94 | 0.18 |
| Parameter | $HQ_c$ for Adults | | $HQ_c$ for Children | |
| | Cd | Pb | Cd | Pb |
| Minimum | $4.8*10^{-6}$ | $3.13*10^{-5}$ | $2.9*10^{-5}$ | $1.67*10^{-9}$ |
| Maximum | $1.2*10^{-4}$ | $14.3*10^{-5}$ | $7.11*10^{-4}$ | $86.1*10^{-5}$ |
| Average | $5.66*10^{-5}$ | $8.41*10^{-5}$ | $3.42*10^{-4}$ | $48.5*10^{-5}$ |